# A Dataset of Visible Light and Thermal Infrared Images for Health Monitoring of Caged Laying Hens in Large-Scale Farming

**DOI:** 10.3390/s24196385

**Published:** 2024-10-02

**Authors:** Weihong Ma, Xingmeng Wang, Xianglong Xue, Mingyu Li, Simon X. Yang, Yuhang Guo, Ronghua Gao, Lepeng Song, Qifeng Li

**Affiliations:** 1Information Technology Research Center, Beijing Academy of Agriculture and Forestry Sciences, Beijing 100097, China; mawh@nercita.org.cn (W.M.); 2022204009@cqust.edu.cn (X.W.); gaorh@nercita.org.cn (R.G.); 2National Innovation Center of Digital Technology in Animal Husbandry, Beijing 100097, China; xuexl@nercita.org.cn (X.X.); limy@nercita.org.cn (M.L.); guoyh@nercita.org.cn (Y.G.); 3College of Electronic and Electrical Engineering, Chongqing University of Science & Technology, Chongqing 401331, China; 4Advanced Robotics and Intelligent Systems Laboratory, School of Engineering, University of Guelph, Guelph, ON N1G 2W1, Canada; syang@uoguelph.ca

**Keywords:** chicken head detection, laying hen counting, caged henhouse health inspection, visible light and thermal infrared image alignment, deep learning

## Abstract

Considering animal welfare, the free-range laying hen farming model is increasingly gaining attention. However, in some countries, large-scale farming still relies on the cage-rearing model, making the focus on the welfare of caged laying hens equally important. To evaluate the health status of caged laying hens, a dataset comprising visible light and thermal infrared images was established for analyses, including morphological, thermographic, comb, and behavioral assessments, enabling a comprehensive evaluation of the hens’ health, behavior, and population counts. To address the issue of insufficient data samples in the health detection process for individual and group hens, a dataset named BClayinghens was constructed containing 61,133 images of visible light and thermal infrared images. The BClayinghens dataset was completed using three types of devices: smartphones, visible light cameras, and infrared thermal cameras. All thermal infrared images correspond to visible light images and have achieved positional alignment through coordinate correction. Additionally, the visible light images were annotated with chicken head labels, obtaining 63,693 chicken head labels, which can be directly used for training deep learning models for chicken head object detection and combined with corresponding thermal infrared data to analyze the temperature of the chicken heads. To enable the constructed deep-learning object detection and recognition models to adapt to different breeding environments, various data enhancement methods such as rotation, shearing, color enhancement, and noise addition were used for image processing. The BClayinghens dataset is important for applying visible light images and corresponding thermal infrared images in the health detection, behavioral analysis, and counting of caged laying hens under large-scale farming.

## 1. Introduction

In the rapidly advancing field of precision livestock farming technology, intelligent management has become a key factor in enhancing production efficiency and reducing costs. Particularly within the poultry farming industry, as the global demand for chicken meat and eggs increases, numerous farms are progressively transitioning towards intelligent and automated systems [1]. Many large-scale poultry farms have developed and implemented intelligent management systems for tasks such as overall temperature control in the henhouses, automated egg collection, and automated feeding, significantly reducing costs. However, many practical issues remain unresolved in the refined, intelligent management of individual hens and cages.

Geffen et al. [2] utilized a convolutional neural network based on Faster R-CNN to detect and count laying hens in battery cages, achieving an accuracy of 89.6%. Bakar et al. [3] demonstrated a novel feature for detecting bacteria- or virus-infected chickens by automatically detecting the color of the chicken’s comb. Yu et al. [4] proposed a cage-gate removal algorithm based on big data and deep learning Cyclic Consistent Migration Neural Network (CCMNN) for the detection of head features in caged laying hens. Wu et al. [5] employed super-resolution fusion optimization methods for multi-object chicken detection, enriching the pixel features of chickens. Campbell et al. [6] utilized a computer vision system to monitor broiler chickens on farms, analyzing the activity of the broilers through an unsupervised 2D trajectory clustering method.

Establishing and refining a refined, intelligent management system for poultry farms are inseparable from the support of high-quality data. The quality and quantity of the dataset play a crucial role in the training of deep learning-based algorithms, the validation of models, and the enhancement of system performance [7]. Yao et al. [8] constructed a gender classification dataset for free-range chickens, which includes 800 images of chicken flocks and 1000 images of individual chickens. Adebayo et al. [9] collected a dataset of 346 audio signals for the health monitoring of chickens based on vocalization signals. Aworinde et al. [10] established a chicken feces image dataset with data labels containing 14,618 images for predicting the health status of chickens. We have yet to encounter a high-quality dataset of visible light and thermal infrared images for caged laying hens.

The primary aim of this paper is to introduce and elaborate on the construction and application of the BClayinghens dataset. The BClayinghens dataset contains visible light and thermal infrared images of caged laying hens, aiming to enhance the detection capabilities of health status, behavior analysis, and counting of caged laying hens through deep learning technologies. This study will provide valuable data resources for researchers and practitioners in the field of caged laying hen research. The specific objectives include the following:(1)Dataset Introduction: A comprehensive description of the background, data content, collection process, data processing, and annotation methods involved in constructing the BClayinghens dataset.(2)Dataset Applications: An exploration of the practical application value of the BClayinghens dataset in chicken head target detection.(3)Deep Learning Model Training: An explanation of how to use this dataset to train deep learning models and enhance the model’s robustness in different environments through data augmentation techniques.(4)Providing Data Resources: Offering high-quality image data resources of caged laying hens to researchers and practitioners, supporting the research and development of poultry health detection and intelligent farming management systems.

## 2. Materials and Methods

### 2.1. Data Collection

Before data collection, we devised a detailed photography plan, aiming to meet the diverse characteristics of the data.

#### 2.1.1. Experimental Location

China stands as one of the largest global producers and consumers of eggs, rendering the collection of laying hen data from Chinese large-scale farms highly significant for investigating the utilization of artificial intelligence technologies in the health monitoring and behavioral recognition of laying hens. We collected data from a large poultry farm in Shenzhou, Hebei, China, where the number of chickens exceeds one million. The collected data pertain to white-feathered laying hens. The housing structure within a single shed consists of a 10-column by 8-tier cage configuration, with partitions every 4 tiers. Each cage accommodates 6 to 8 laying hens, with an age range of 200 to 300 days. The dimensions of an individual cage are 60 cm in length and 60 cm in width, with a front height of 50 cm and a rear height of 38 cm, and the cage floor is inclined at approximately 10°. The data collection apparatus is integrated into the Autonomous Poultry House Inspection Robot, maintaining a distance of 60 cm from the cage’s exterior during its patrol for mobile monitoring purposes.

#### 2.1.2. Experimental Equipment

To facilitate data collection, this study explored the use of an autonomous inspection robot for poultry houses, as shown in Figure 1 [11]. The technical parameters of the robot are detailed in Table 1. The autonomous inspection robot can adjust its speed, direction, image acquisition, and detection height according to the specific tasks of the experiment, and it provides a visual interface as well as monitoring indoor gas, temperature, and humidity information.

Data collection devices consist of smartphones and chicken inspection robots equipped with visible light cameras and infrared thermal cameras. The specific parameters of the image acquisition devices are detailed in Table 2. Given the varying image qualities produced by the three types of collection devices, the BClayinghens dataset adeptly captures the diversity of images within the henhouse environment, as depicted in Figure 2.

Notably, during the acquisition of TIR images by the chicken inspection robot, the camera is simultaneously activated to capture RGB images (with approximately a 1 s interval between the two image captures). Consequently, each TIR image of the henhouse has a corresponding RGB image, providing viable data material for the design of image fusion algorithms [13]. The chicken inspection robot and its image capture devices are shown in Figure 3. Additionally, by scaling and shifting the RGB images and conducting boundary checks and dimension adjustments, we have achieved coordinate correction between RGB and TIR images.

#### 2.1.3. Experimental Methods

In large-scale poultry farms, the internal environment of the henhouses is uniformly regulated by a central environmental control system. The main factors affecting the experimental data include the following: resting hens, feed cart feeding, egg laying by laying hens, and pecking behaviors of the chickens. Under a single experiment, the chicken dataset is captured in 12 sessions; each shooting session lasted 2 h, as shown in Table 3.

The data collection experimental process is shown in Figure 4. After the poultry house inspection robot is activated, the lifting device equipped with a thermal infrared camera and a visible light camera automatically rises to the predetermined height according to the structure of the poultry house and completes the camera initialization. Subsequently, the robot’s position-reading device reads the cage number while the thermal infrared camera and visible light camera automatically capture images of the hens, continuing until the last cage is completed. The thermal infrared camera and visible light camera are centrally controlled by the information collection system, completing one round of data collection by scanning around 960 cages to obtain diverse data. After the final hen image is captured, the lifting device, thermal infrared camera, and visible light camera reset, and all data are archived by the robot and sent to the cloud.

It should be noted that the data collection experiments described in this paper used visible light cameras and thermal infrared cameras that employ non-stress, non-contact image acquisition methods. Additionally, the farm enforced strict disinfection and protection measures for personnel entering the site, ensuring that the image collection and testing process did not cause any harm to the hens. Therefore, the research presented in this paper does not raise any ethical issues. All images were captured between April 2022 and July 2024, with image acquisition experiments in five coops in separate time periods for a total of 60 shots. This resulted in 30,993 raw images with a data file size of 6.89 GB.

#### 2.1.4. Data Preprocessing

Data preprocessing encompasses three aspects. 1: For a series containing only RGB images, data cleansing is required. Duplicate and chicken-free images are removed to meet the needs of chicken head detection tasks and to facilitate the exploration of performance enhancement in object detection by deep learning algorithms. 2: For a series of images where RGB corresponds to TIR images, the correspondence between RGB and TIR images is verified, and any RGB and TIR images not containing laying hens are removed. 3: The selected data are annotated to distinguish between the chicken head and body parts, which will be beneficial for implementing chicken counting and dead chicken detection tasks. The data annotation work is completed by data annotators, ensuring the accuracy of the data labels.

To ensure the quality of the dataset, data selection was performed manually. The data selection criteria are detailed in Table 4.

### 2.2. Data Augmentations

The limitations of the poultry farm environment can result in a singular environment within the collected laying hen image dataset and a limited number of collections. To better adapt this dataset to various breeding environments and deep learning models, we have employed data augmentation techniques to enrich the existing data. Data augmentation considers the impact of factors such as the breeding environment, collection equipment, and chicken head features on the generalizability of the dataset, making the augmented dataset more accurate and reliable in model expression.

#### 2.2.1. Rotation

Three rotation methods are used for data augmentation: rotating the image’s red channel by 45°, shearing the image in a 16° direction to create a tilt effect, and rotating the image counterclockwise by 20°. Rotation augmentation is primarily used for enhancing RGB-TIR images, and the enhanced RGB images remain corresponding to the TIR images.

#### 2.2.2. R-Value (Redness) Enhancement

The key to accurately identifying laying hens is precisely determining the chicken head. Since the comb color of a laying hen is red, enhancing the value of the red channel in the RGB image will facilitate the differentiation between the target and the background. We randomly increase the R channel of each pixel of the image by a value between 10 and 100, as follows:(1)$$R_{enhanced}\left(x,y\right)=R_{raw}\left(x,y\right)+\Delta R$$
where *R_enhanced_* is the red channel value of the pixel at position (*x*, *y*) in the enhanced image; Δ*R* is a random variable ranging from [10, 100]; *R_raw_* is the red channel value of the pixel at position (*x*, *y*) in the raw image.

#### 2.2.3. Hue Enhancement

Hue enhancement can make the model more robust during training and capable of dealing with the impact of hue variation. First, convert the image from the RGB color space to the HSV color space. Then, select the hue channel H and add a random value. The hue enhancement is calculated as follows:(2)$$H_{enhanced}=H_{raw}+\Delta H$$
where *H_enhanced_* is the enhanced *H* channel value; Δ*H* is a random variable ranging from [25, 90]; *H_raw_* is the H channel value of the raw image. Since hue *H* is periodic (between 0 and 360 degrees), periodic adjustment is made to *H_enhanced_* during the enhancement process to ensure it remains within the valid hue range.

#### 2.2.4. Brightness Enhancement

Brightness enhancement is used to simulate lighting variations in different poultry farms, variations in lighting at different heights of chicken cages within the same farm, and imaging differences in image collection by different cameras, thereby increasing data complexity. A multiplication factor is applied to each channel of every pixel in the image, calculated as follows:(3)$$P_{enhanced}=k\times P_{raw}$$
where *P_enhanced_* is the pixel value of the enhanced image; *k* is a random variable ranging from [0.5, 1.5]; *P_raw_* is the pixel value of the raw image. When *k* is less than 1, the enhanced image is darker than the raw image; when *k* is greater than 1, the enhanced image is brighter than the raw image.

#### 2.2.5. Edge Enhancement

Edge enhancement is used to make the edges of the image more pronounced, which can improve the image’s visual effect [14]. Incorporating edge enhancement as a step in the laying hen data augmentation process during machine learning model training can enhance the model’s ability to recognize edge features of the chicken head (especially the comb part).

#### 2.2.6. Gaussian Noise

Gaussian noise is added for data augmentation to simulate noise interference during the image collection process in the henhouse. A random noise value following a Gaussian distribution is added to each raw image pixel. The enhanced pixel value is calculated as follows:(4)$$P_{Gnew}=clip\left(P_{raw}+N,0,255\right)$$
where *P_Gnew_* is the pixel value of the image after Gaussian enhancement; the *clip* function limits the value to the range of 0 to 255; *N* is the noise value, randomly drawn from a Gaussian distribution with a standard deviation of 51.

#### 2.2.7. Laplacian Noise

Laplacian noise enhancement is added to simulate interference caused by low lighting and high compression rates during image collection in the henhouse environment. A random noise value following a Laplacian distribution is generated for each pixel of the image, calculated as follows:(5)$$P_{Lnew}=clip\left(P_{raw}+M,0,255\right)$$
where *P_Lnew_* is the pixel value of the image after Laplacian enhancement; the *clip* function limits the value to the range of 0 to 255; *M* is the noise value, randomly drawn from a Laplacian distribution with a scale of 51.

#### 2.2.8. Poisson Noise

Statistical noise following a Poisson distribution is added to enhance the robustness of the model, calculated as follows:(6)$$P_{Pnew}=clip\left(P_{raw}+\lambda ,0,255\right)$$
where *P_Pnew_* is the pixel value of the image after Poisson enhancement; the *clip* function limits the value to the range of 0 to 255; *λ* is the noise value, randomly drawn from a Poisson distribution with a scale of 40.

#### 2.2.9. Salt-and-Pepper Noise

Salt-and-pepper noise is added for data augmentation to simulate interference in images due to low-quality cameras, wireless transmission errors, storage media damage, and other similar issues. The proportion of pixels covered by noise points in the image is defined as 10%, and within this 10%, white (salt) or black (pepper) pixel points are randomly introduced.

The raw images were processed using the methods above, as illustrated in Figure 5. Before data augmentation, there were 30,993 raw images, with 2359 xml, json, and txt data labels each. It is important to note that the laying hen data obtained through the above data augmentation methods include not only the laying hen images but also the corresponding xml, json, and txt data labels for the augmented laying hen images, or the TIR images corresponding to the RGB images. This allows for the support of various model training tasks. After data augmentation, there are 61,133 images and 63,693 data labels (21,231 XML, JSON, and TXT data labels each).

## 3. Results and Analysis

### 3.1. Data Description

This dataset encompasses three categories of imagery data about laying hens: RGB image data, paired RGB and Thermal Infrared (TIR) image data, and RGB image data incorporating chicken head annotation information. Specifically, the dataset comprises 5226 RGB images of individual hens, constituting 8% of the total dataset. The paired RGB and TIR images number 34,676, representing 57% of the dataset. The RGB images annotated with chicken head information total 21,231, accounting for 35% of the dataset.

#### 3.1.1. Data Environment

A robust dataset must consider both the quantity and quality of the images and the diversity of the environment [15], taking into account a variety of photographic perspectives, lighting conditions, and background contexts. Figure 6, Figure 7 and Figure 8 depict the range of complex environments captured within the dataset.

(1)We collected data on individual laying hens in a natural environment, as illustrated in Figure 6. During photography, complete and partial images of the hens were captured under various natural lighting and background conditions. The data obtained from the natural setting are conducive to analyzing the morphology of free-range laying hens [16] and assessing their health state.(2)The activity states of caged laying hens were photographed, considering the variations in cage lighting and camera angles, as depicted in Figure 7. The dataset of caged laying hens can be utilized to analyze flock activity levels, detect deceased hens within cages, and calculate hen population numbers.(3)An innovative image acquisition technology that fuses RGB images with TIR images was introduced [17], as shown in Figure 8. Corresponding RGB and TIR images were taken at different distances, angles, and with varying numbers of hens, capturing multimodal image data. The image fusion technique overcomes interference caused by camera shake in RGB images and provides data support for studying infectious diseases among hen flocks.

Due to the slightly wider field of view in TIR image compared to RGB images (with the TIR image resolution being lower than that of the RGB images), we employed image correction [18] methods to transform the coordinates and dimensions of the RGB images onto the TIR images to align these two kinds of images. Using the *xywh* notation for image correction, the calculation method for the two-dimensional coordinates of pixels and the dimensions of length and width in the TIR image is as follows:(7)$$\left\{\begin{array}{l} T_{x}=k_{x}\cdot x+b_{x}\\ T_{y}=k_{y}\cdot y+b_{y}\\ T_{w}=k_{w}\cdot w\\ T_{h}=k_{h}\cdot h \end{array}\right.$$
where *x* represents the *x*-coordinate in the RGB image with the top-left corner as the origin; *y* represents the *y*-coordinate in the RGB image with the top-left corner as the origin; *w* denotes the raw width of the RGB image; *h* denotes the raw height of the RGB image; *T_x_* signifies the *x*-coordinate in the transformed TIR image; *T_y_* signifies the *y*-coordinate in the transformed TIR image; *T_w_* denotes the width of the transformed TIR image; *T_h_* denotes the height of the transformed TIR image; *k_x_* is the scaling factor for the *x*-coordinate, taken as 0.09661 in this paper; *k_y_* is the scaling factor for the *y*-coordinate, taken as 0.09558 in this paper; *k_w_* is the scaling factor for the image width, taken as 0.09672 in this paper; *k_h_* is the scaling factor for the image height, taken as 0.09618 in this paper; *b_x_* is the adjustment factor for the *x*-coordinate offset, taken as 8.66434 in this paper; *b_y_* is the adjustment factor for the *y*-coordinate offset, taken as 10.1642 in this paper.

To ensure that the transformed coordinates and dimensions do not exceed the boundaries of the TIR image, boundary checks and dimension adjustments are performed on the corrected TIR images. The adjusted width *T_adjust_*_(*w*)_ of the TIR image is calculated as follows:(8)$$T_{adjust(w)}=\left\{\begin{array}{l} T_{w}\;\;\;if\left(T_{x}+T_{w}\leq 318\right)\\ 318-T_{x}if\left(T_{x}+T_{w}>318\right) \end{array}\right.$$

The adjusted height *T_adjust_*_(*h*)_ of the TIR image is calculated as follows:(9)$$T_{adjust(h)}=\left\{\begin{array}{l} T_{h}\;\;\;if\left(T_{y}+T_{h}\leq 238\right)\\ 238-T_{y}if\left(T_{y}+T_{h}>238\right) \end{array}\right.$$

Furthermore, data augmentation techniques have been applied to the dataset to expand it, including operations such as rotation, hue transformation, brightness transformation, and the addition of noise.

#### 3.1.2. Annotations and Labels

In the collected images of caged laying hen flocks, annotations for chicken heads were performed, with the data annotation task accomplished by 10 data annotators, ensuring the precision and standardization of the data. The annotations provide location information for the heads of the hens within the images, marked with rectangular frames, using the Labelme (Version 4.5.13) annotation software. To ensure the quality and consistency of the annotations, the following methods were used for data quality inspection. 1. After completing the data annotation, the 10 annotators conducted cross-checks in pairs to ensure that there were no instances of missed annotations, incorrect annotations, or inaccurate annotation positions for the chicken heads. 2. Researchers with interdisciplinary expertise in animal husbandry management and information technology, along with poultry farm staff, jointly verified whether the categories and positions of the data labels were correct.

As shown in Figure 9, for the same image, we provide annotation information in three different formats: xml, txt, and json. The xml format annotation files include the images’ size, category, and location information, adhering to the annotation standards of datasets such as PASCAL VOC and YOLO. The txt format is a plain text file containing only category and location information, commonly used for YOLO format annotations, and is characterized by its simplicity and ease of parsing. The json format annotation files include image size, category, location, and other information, with a level of lightness between xml and txt formats. The three data annotation formats we provide can meet the needs of different application scenarios. For the operation guide on data annotation, please refer to https://github.com/maweihong/BClayinghens/blob/main/Data%20Annotation%20Guide.docx (accessed on 1 September 2024).

#### 3.1.3. Data Folders

The structure of the data folders is shown in Figure 10 and Figure 11. In Figure 11, the folders from 09RBGtxt+xml+json-2359 to 17SPNoisetxt+xml+json-2359 have divided the dataset in a ratio of 7:2:1, with the training set (folder train1653) accounting for 70%, the validation set (folder val471) accounting for 20%, and the test set (folder test235) accounting for 10%. Each folder for the training, validation, and test sets contains four subfolders: img, json, txt, and xml, for saving the corresponding formats of ten images or labels. Figure 12 and Figure 13 show the file count statistics within the BClayinghens dataset.

#### 3.1.4. Value of the Data

The dataset is applicable for the health monitoring and behavioral analysis of caged laying hens and research on chicken head detection and the counting of caged hens. It is particularly suitable for classification studies of hens based on deep learning convolutional neural networks [19].Multimodal image data from visible light and thermal infrared are provided, supporting research on the health monitoring and behavior recognition of hens through image fusion techniques. The fused images of hens are especially useful for body temperature detection, aiding disease prevention and control efforts in commercial poultry farming.Different morphological images of individual hens are supplied, which can be utilized for health assessments of laying hens.The use of the BClayinghens dataset for automatic monitoring and management [20] in laying hen farming not only enables the detection of disease risks in hens under non-contact, stress-free conditions but also improves the efficiency of routine inspections and the accuracy of disease detection [21], while reducing the labor costs of manual inspections in poultry farms. In contrast, the current health management methods for caged laying hens in commercial farms involve manually wearing headlamps and masks to inspect hens row by row. Such intense lighting and human contact not only negatively impact hen welfare but also pose potential health risks to on-site personnel.

The relevant information regarding the BClayinghens dataset is presented in Table 5.

### 3.2. Chicken Head and Head Temperature Detection

#### 3.2.1. Comparison of Detection Performance of Different Algorithms

The chicken inspection robot in the henhouse should possess high real-time performance during actual use, necessitating the selection of target detection models with superior real-time capabilities for deployment [22]. We have chosen mainstream and novel real-time target detection models to test the dataset, thereby verifying the dataset’s outstanding performance across various target detection algorithms. The RT-DETR model [23] and the YOLO series target detection models (YOLOv5 [24], YOLOv8 [25], YOLOv9 [26], YOLOv10 [27]) are utilized as baseline testing methods for chicken head recognition. Model training is implemented on a Linux server, using an Intel(R) Xeon(R) Platinum 8375c CPU @ 2.90GHz processor(Intel, Hillsboro, OR, USA), an NVIDIA RTX A6000 graphics card(NVIDIA, Taiwan, China), CUDA version 11.3, Python version 3.8.18, and the Pytorch deep learning framework, Pytorch version 1.12.0. None of the experiments uses pre-trained weights, with a batch size set to 48 and the number of epochs set to 200.

We selected 2359 images from the BClayinghens dataset to verify the chicken head detection performance of the dataset across different algorithms. Figure 14 illustrates the distribution of the dataset used during the chicken head detection training process, where the coordinates are normalized values. The x and y coordinates, with a value of 0, represent the upper left corner of the image in the dataset, and a value of 1 represents the upper right corner. It can be observed that the x-coordinates of the chicken head target boxes in the selected data are relatively evenly distributed, the y-coordinates are concentrated in the central area of the image, and the width and height of the target boxes follow a normal distribution. Additionally, a denser concentration of points is shown in areas with smaller widths and heights within the image, indicating a strong commonality in certain specific sizes or positions of the chicken head targets in the dataset. This aggregation can help the model learn the target position and size distribution more accurately, thereby improving detection accuracy.

Table 6 presents the detection performance of various algorithms in the chicken head detection task. As the model size, parameters, and computation decrease, the inference speed of the model increases and resource consumption decreases. Among the models, YOLOv5n, YOLOv8n, YOLOv9t, and YOLOv10n exhibit faster inference speeds and lower resource consumption. Conversely, YOLOv5x, YOLOv8x, and YOLOv9e perform poorly in the aforementioned three performance metrics. Therefore, for deploying chicken head detection models on mobile or embedded devices, YOLOv5n, YOLOv8n, YOLOv9t, and YOLOv10n should be considered as preliminary options.

However, in practical applications, it is necessary to balance the model’s Precision, Recall, mAP50, and other accuracy metrics in addition to considering computational resources. As shown in Table 6, the four models with the highest Precision are YOLOv5m, YOLOv8m, YOLOv8l, and YOLOv8x, with Precision values of 96.27%, 96.48%, 95.98%, and 96.28%, respectively. The two models with the lowest Precision are YOLOv9t and YOLOv10n, with Precision values of 92.24% and 91.87%, respectively. Higher Precision indicates more accurate classification of chicken heads, so YOLOv5m, YOLOv8m, YOLOv8l, and YOLOv8x have better classification accuracy. The four models with the higher Recall are YOLOv5s, YOLOv5x, YOLOv8l, and YOLOv8x, with Recall values of 95.82%, 95.23%, 95.54%, and 95.73%, respectively. The two models with lower Recall are RT-DETR-ResNet101 and YOLOv9t, with Recall values of 92.18% and 92.12%, respectively. Higher Recall indicates more accurate localization of chicken heads, so YOLOv5s, YOLOv5x, YOLOv8l, and YOLOv8x offer better localization accuracy. The four models with the higher mAP50 are YOLOv8m, YOLOv9m, YOLOv9c, and YOLOv9e, with mAP50 values of 97.71%, 97.92%, 97.81%, and 97.87%, respectively. The two models with lower mAP50 are RT-DETR-ResNet101 and YOLOv10n, both with an mAP50 value of 95.94%. Higher mAP50 indicates better overall performance of the chicken head detection algorithm.

After considering the performance of the models based on overall metrics—namely inference speed, resource consumption, and accuracy—across a total of six indicators, we recommend the models that ranked in the top positions for at least two of these metrics. Consequently, the preferred chicken head detection models are YOLOv5n, YOLOv8m, and YOLOv9c. Furthermore, for deployment on resource-limited mobile or embedded devices, we recommend using the YOLOv5n model for chicken head detection tasks. For hardware with ample computational resources, we suggest deploying the YOLOv8m or YOLOv9c models, as they offer higher detection accuracy.

Figure 15, Figure 16 and Figure 17 illustrate the variations in *loss* values and *precision* during the training process of the RT-DETR and YOLO series chicken head detection models. Given the complexity of object detection tasks, models require optimization across multiple dimensions. To enable models to comprehensively learn various aspects of object detection tasks, different algorithms enhance overall model performance by employing a design philosophy that combines multiple loss functions. Specifically, in Figure 15, Figure 16 and Figure 17, the *loss* calculation for RT-DETR includes *giou_loss* (Generalized Intersection over Union Loss [28]), *cls_loss* (classification loss), and *l1_loss* (Mean Absolute Error) [23]; YOLOv5 comprises *box_loss* (bounding box loss) and *obj_loss* (object classification loss); YOLOv8 includes *box_loss*, *cls_loss*, and *dfl_loss* (distribution focal loss); YOLOv9 consists of *box_loss*, *cls_loss*, and *dfl_loss*; YOLOv10 utilizes a dual label assignment strategy, encompassing *box_om*, *cls_om*, *dfl_om*, *box_oo*, *cls_oo*, and *dfl_oo* [27].

In Figure 15, the loss and precision curves for the RT-DETR series algorithms show near convergence. Among them, the *giou_loss*, *cls_loss*, and *l1_loss* for RT-DETR-ResNet50 converge to their lowest values, reaching 0.29, 0.35, and 0.11, respectively. Conversely, RT-DETR-l exhibits the highest *giou_loss*, *cls_loss*, and *l1_loss*, at 1.73, 0.90, and 0.92, respectively. In terms of precision, RT-DETR-ResNet101 exhibits significant oscillations during the early training phases but stabilizes to 0.92 in the later stages. Although RT-DETR-ResNet50 also experiences considerable oscillation in the early stages, its final precision of 0.94 surpasses that of RT-DETR-l, RT-DETR-ResNet101, and RT-DETR-x. The differences between the loss and precision curves of the various RT-DETR algorithms are minimal, with the maximum loss function difference being 0.24 and the maximum precision difference being 0.02 after convergence. Overall, when applied to the BClayinghens dataset, the RT-DETR-ResNet50 algorithm demonstrates the best classification and regression performance among the RT-DETR series.

In Figure 15, the differences in the loss curves among the YOLOv5 series algorithms are quite pronounced, while the precision curves show smaller variations. YOLOv5l and YOLOv5x both converge well, with YOLOv5x exhibiting the best convergence. Specifically, YOLOv5x achieves the lowest convergence for *box_loss*, reducing from 0.11 to 0.01, and for *obj_loss*, decreasing from 0.08 to 0.02. Among the YOLOv5 series algorithms, YOLOv5m, YOLOv5l, and YOLOv5x all achieve a precision of 0.96. Therefore, when applying the BClayinghens dataset to the YOLOv5 series algorithms, selecting YOLOv5x provides the best trade-off between loss and precision.

Figure 16 illustrates the variations in loss values and precision for the YOLOv8 and YOLOv9 chicken head detection models. Among the YOLOv8 series models, the largest model, YOLOv8x, achieves the lowest loss values, with *box_loss* converging to 0.31, *cls_loss* converging to 0.18, and *dfl_loss* converging to 0.84. In contrast, the smallest model in the YOLOv8 series, YOLOv8n, has *box_loss*, *cls_loss*, and *dfl_loss* values of 0.84, 0.40, and 1.15, respectively. We observe that the loss curves for the YOLOv8 series algorithms converge better as the model size increases. Additionally, precision exhibits a similar trend, with YOLOv8n having the lowest precision among the YOLOv8 series models, at 0.95.

In Figure 16, the loss curves for the YOLOv9 series algorithms exhibit similar characteristics to those of the YOLOv8 series, with the loss curves converging better as the model size increases. YOLOv9t achieves *box_loss*, *cls_loss*, and *dfl_loss* values of 1.42, 0.67, and 1.63, respectively. In comparison, YOLOv9e achieves *box_loss*, *cls_loss*, and *dfl_loss* values of 0.82, 0.39, and 1.28. In terms of precision, the highest value observed is 0.96, with YOLOv9m, YOLOv9c, and YOLOv9t all reaching this level of precision.

Figure 17 displays the loss and precision curves for the YOLOv10 series algorithms. Both the one-to-many head (om) and one-to-one head (oo) variants of the YOLOv10 series show convergent loss curves. YOLOv10l achieves the lowest values for *box_om*, *cls_om*, and *dfl_om*, which are 0.48, 0.25, and 0.89, respectively. YOLOv10l also has the lowest values for *box_oo*, *cls_oo*, and *dfl_oo* among the YOLOv10 series algorithms, with values of 0.50, 0.18, and 0.90. Additionally, YOLOv10l achieves the highest precision in this series, at 0.96. Therefore, when applying the BClayinghens dataset with the YOLOv10 series algorithms, using YOLOv10l will yield better detection precision.

Overall, as the number of training epochs increases, the loss values for each algorithm gradually decrease and stabilize, while the precision values gradually increase and stabilize. The highest convergence of the loss values among the algorithms reaches 0.02. For detection tasks where precision is prioritized, selecting one of the following seven algorithms—YOLOv5m, YOLOv5l, YOLOv5x, YOLOv9m, YOLOv9c, YOLOv9t, or YOLOv10l—can achieve the highest precision of 0.96.

Additionally, this study evaluates the detection performance of each model using Recall and mAP50 (mean Average Precision at Intersection over Union of 0.5), as shown in Figure 18. It can be observed that, with the exception of the RT-DETR-ResNet50 and YOLOv5x algorithms, the Recall and mAP50 values of the other algorithms converge rapidly in the early stages of training and exhibit minimal oscillation.

As shown in Figure 18, RT-DETR-ResNet50 has the highest Recall in the RT-DETR series, at 0.94, while RT-DETR-x achieves the highest mAP50 in the RT-DETR series, at 0.97. Given that Figure 15 indicates RT-DETR-ResNet50 performs best in classification and regression with the highest Recall and an mAP50 of 0.96, we determine that the best-performing algorithm for the BClayinghens dataset within the RT-DETR series is RT-DETR-ResNet50.

As shown in Figure 18, within the YOLOv5 series algorithms, Recall is highest for YOLOv5s, at 0.96. The highest mAP50 is 0.98, achieved by YOLOv5n, YOLOv5s, and YOLOv5m. The differences between YOLOv5 algorithms after convergence are not significant. When training YOLOv5 series algorithms on the BClayinghens dataset, if considering a balance between loss and precision, YOLOv5x is recommended. For a focus on high Recall and high mAP50, YOLOv5s should be selected.

In the YOLOv8 series algorithms, Recall and mAP50 show significant differences during the first 75 epochs of training, but converge closely in the later stages of training. Recall ranges from 0.94 to 0.95, while mAP50 ranges from 0.97 to 0.98. The BClayinghens dataset demonstrates good adaptability within the YOLOv8 series algorithms.

In the YOLOv9 series algorithms, Recall reaches its highest value of 0.96 with the YOLOv9e model, while the YOLOv9t model shows the lowest Recall of 0.92 within the series. mAP50 is 0.98 for the YOLOv9m, YOLOv9c, and YOLOv9e algorithms, and 0.97 for the YOLOv9t and YOLOv9s algorithms. While the differences in mAP50 among YOLOv9 series algorithms are not significant, YOLOv9e exhibits the highest values for loss, Recall, and mAP50 within this category. Therefore, the best-performing algorithm for the BClayinghens dataset in the YOLOv9 series is determined to be YOLOv9e.

In the YOLOv10 series algorithms, Recall reaches its maximum value of 0.94 with the YOLOv10s and YOLOv10l models. mAP50 achieves its highest value of 0.97 with the YOLOv10s, YOLOv10b, YOLOv10l, and YOLOv10x models. Based on the YOLOv10 loss and precision curves shown in Figure 17, the YOLOv10l model performs optimally across various aspects. Therefore, the best-performing algorithm for the BClayinghens dataset in the YOLOv10 series is determined to be YOLOv10l.

It can be observed that the dataset exhibits low loss values and high precision, recall, and mAP50 values across different models, demonstrating a strong capability to identify chicken head targets. Among all algorithms, the highest loss value converges to 0.02, the highest precision is 0.96, the highest recall is 0.96, and the highest mAP50 is 0.98. Overall, the BClayinghens dataset has high quality and can adapt well to various object detection algorithms.

#### 3.2.2. Analysis of Chicken Head Detection

Due to the interpretability limitations of deep learning algorithms, we conducted a visual analysis using the inference results of chicken head detection. Figure 19, Figure 20, Figure 21, Figure 22 and Figure 23 demonstrate the recognition effects of different object detection algorithms on laying hen images. In Figure 19, the highest confidence for chicken head detection by the RT-DETR series algorithms is 0.95. In Figure 20, the highest confidence for chicken head detection by the YOLOv5 series algorithms is 0.98. In Figure 21, the highest confidence for chicken head detection by the YOLOv8 series algorithms is 0.98. In Figure 22, the highest confidence for chicken head detection by the YOLOv9 series algorithms is 0.98. In Figure 23, the highest confidence for chicken head detection by the YOLOv10 series algorithms is 0.99.

In general, all algorithms can accurately identify the chicken head targets for images of laying hens in various poses with high confidence. The predicted boxes can fully encompass the chicken head area without any missed detections. The RT-DETR series algorithms have chicken head detection confidence, mostly around 0.9, with relatively stable detection results. The YOLOv5 series algorithms have the highest sensitivity in identifying chicken heads, with detection confidence frequently reaching up to 0.98. The YOLOv5 series algorithms have the best learning capability for chicken head features but are prone to false positives in practical applications. The YOLOv8 series algorithms have the highest confidence at 0.97, and the YOLOv9 series algorithms have the highest confidence at 0.98. However, considering the *loss* values and *precision* curves during the training process, we found that the YOLOv8 series algorithms consistently have lower *loss* values than YOLOv9, indicating that the YOLOv8 series algorithms have an advantage in chicken head detection. In comparison, the YOLOv10 series algorithms are less capable of recognizing chicken head targets than others, with the lowest confidence for single chicken head recognition at 0.3. In conclusion, different mainstream object detection models can meet the needs of chicken head identification and provide great convenience for subsequent chicken head temperature detection and deployment on the robot side.

#### 3.2.3. Analysis of Chicken Head Temperature Identification

After applying the object detection algorithm to identify the chicken head using the chicken inspection robot in the henhouse, the position of the chicken head is sent to the information collection and control system in the *xywh* (coordinates (*x*, y), width *w*, height *h*) format, as shown in Equations (7)–(9). The information collection and control system then invokes the chicken head temperature recognition algorithm and displays the chicken head temperature recognition results on the robot’s visualization interface.

During the experimental process, we calibrated the temperature recognition system of the chicken inspection robot in the henhouse to further improve the temperature detection accuracy. The method for correcting the chicken head temperature detection is as follows:*T_a_* = 0.144 · *T_t_* + 36.351(10)
where *T_t_* is the raw chicken head temperature detected by the infrared thermal camera, and *T_a_* is the corrected actual chicken head temperature.

When the detection and recognition results of the chicken head are accurate, the chicken inspection robot in the henhouse can call the temperature detection algorithm to calculate the chicken head temperature. Since the RT-DETR above and YOLO series object detection algorithms can accurately identify the chicken head targets, the effect of chicken head temperature recognition is only related to the temperature recognition system. The YOLOv5n algorithm, which has the highest sensitivity for chicken head detection in the YOLOv5 series, was deployed on the chicken inspection robot in the henhouse to test the effect of chicken head temperature detection. The YOLOv5n algorithm can maintain a fast speed while providing accuracy suitable for edge devices, and the effect of chicken head temperature recognition is shown in Figure 24.

In Figure 22, the chicken head temperature is generally around 40 °C, the normal temperature for a chicken head, indicating a high degree of temperature recognition accuracy. Moreover, the movement of the chicken and the robot has a minimal impact on the recognition of the chicken head temperature, with the maximum difference in the temperature recognition of the same laying hen’s head being 0.4 °C and the minimum being 0.1 °C. The dataset shows good performance in chicken head temperature recognition, which can provide a strong preference for the refined management and prevention of diseases in the henhouse.

#### 3.2.4. Comparison with Existing Research

This section discusses the advantages and limitations of this study compared to existing research in terms of dataset quality, chicken head detection accuracy, and chicken head temperature detection accuracy.

In terms of dataset, Reference [8] provides a free-range chicken gender classification dataset with a size of 355 MB. In contrast, the BClayinghens dataset provided in this study includes 61,133 images of caged laying hens, with a dataset size of 76.1 GB, offering more extensive data. For the 346 audio signal dataset provided in Reference [9], the authors did not apply any algorithms for data usability validation. In contrast, our study, which tested 25 different algorithms, demonstrates that the BClayinghens dataset achieves over 95% accuracy in chicken head detection tasks. Reference [10] provides a chicken manure dataset containing 14,618 images, each with a size of 100 × 100 pixels. In comparison, the BClayinghens dataset offers better and richer image quality with resolutions of 2592 × 1944, 1920 × 1080, and 320 × 240.

In terms of chicken head detection, Reference [29] employs adaptive brightness adjustment and convolutional neural networks for the fine detection of caged hens’ head states, achieving a mAP of 0.846. Reference [4] uses YOLOv8n, YOLOv8s, YOLOv8m, and YOLOv8x for caged chicken head feature recognition, with the highest mAP value reaching 0.988. In this study, YOLOv8 series algorithms were also used for chicken head detection, with the highest recognition mAP being 0.98, comparable to the mAP in Reference [4]. In addition to YOLOv8 algorithms, this study utilized a total of 25 algorithms from five series including RT-DETR, YOLOv5, YOLOv9, and YOLOv10 for caged laying hen head detection. The results indicate that the highest precision for chicken head detection is 0.96, recall is 0.96, and mAP50 is 0.98.

In terms of chicken head temperature detection, References [30,31] estimate the sensible heat loss (SHL) of laying hens using infrared thermal imaging (TIR), showing that the temperature of the chicken head ranges between 32.9 °C and 42.1 °C. References [30,31] do not provide the average temperature of the chicken head, but from the TIR images in the references, it can be observed that only the eye area of the chicken head has a lower temperature; thus, it is inferred that the average temperature of the chicken head is above 37 °C. The average temperature of the chicken head measured in this study ranges from 40.7 °C to 41.6 °C, which is consistent with the experimental results of References [30,31] and this study provides a larger sample of chicken head temperature data. Reference [32] proposes a temperature measurement method for pig eyes, with an average temperature of 32.34 °C and an error of 1.06 °C for pigs with normal body temperature, and an average temperature of 34.37 °C with an error of 0.41 °C for pigs with abnormal body temperature. The temperature measurement error for chicken heads in this study ranges from 0.1 °C to 0.4 °C, which is smaller than the error reported in Reference [32].

Overall, this study utilizes the BClayinghens dataset for chicken head detection and temperature measurement. Compared to existing research, the dataset in this study offers better image quality, the chicken head detection task achieves higher precision, and the chicken head temperature measurements have a lower rate of errors.

## 4. Conclusions

In this study, we have established a comprehensive image dataset named BClayinghens, which encompasses RGB and TIR images of flocks within henhouses and RGB images annotated with laying hen head information. The dataset includes images captured under diverse conditions, varying scenarios, sizes, and lighting types, making it suitable for various image detection tasks for laying hens and for research in deep learning methods within the caged laying hen production domain. Specifically, it includes (1) RGB images of individual laying hens that can be utilized to evaluate their size, thereby facilitating the calculation of weight and the degree of growth in individual parts; it can also be employed to assess the health status of the hens and to identify deceased or sickly hens. (2) RGB and TIR images of flocks within cages are conducive to monitoring individual and group temperatures, aiding in preventing and controlling diseases within hen populations; they are also useful for detecting deceased or sickly hens in complex cage environments. (3) RGB images with annotated laying hen head information can be applied for counting caged laying hens, simplifying intelligent management for large-scale egg-production facilities. The BClayinghens dataset offers significant contributions to the study of animal welfare for caged laying hens, propelling the advancement of smart henhouse management and enhancing the welfare of the animals.

## Figures and Tables

**Figure 1 sensors-24-06385-f001:**
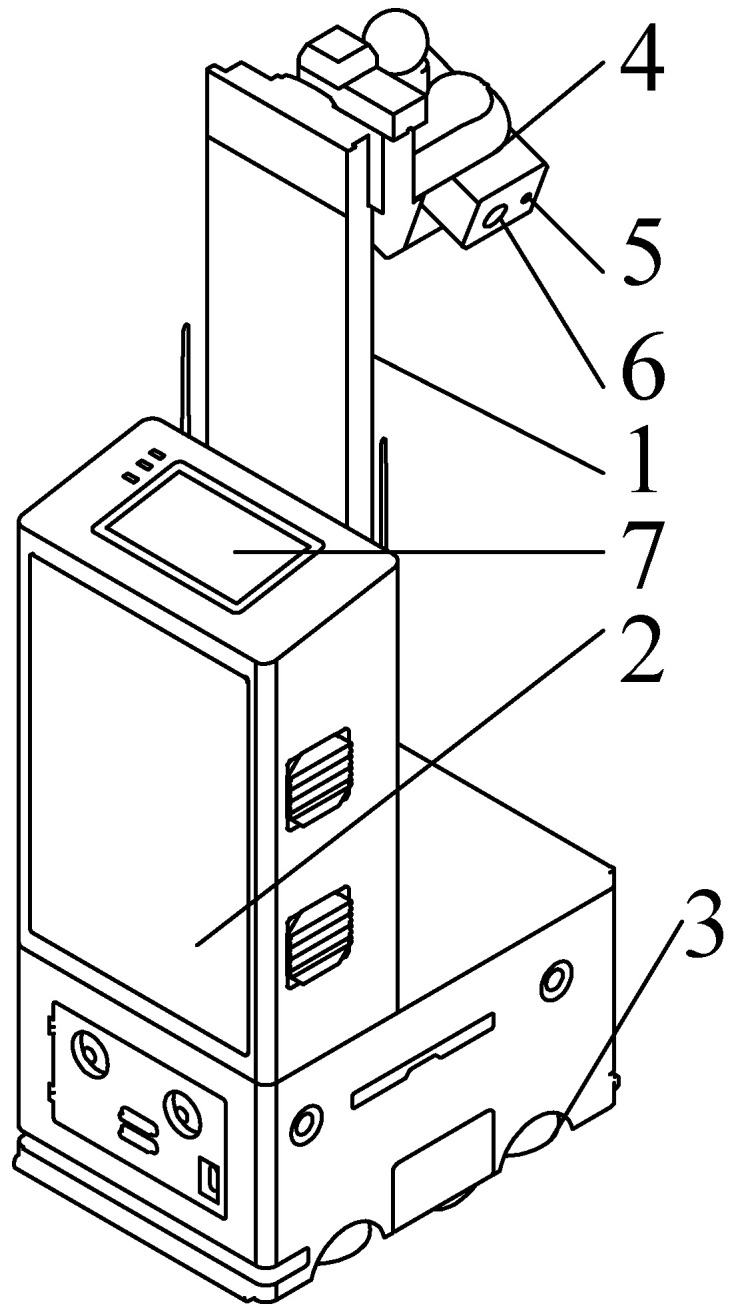
Data collection device—autonomous poultry house inspection robot. 1, Lifting device of the chicken inspection robot; 2, information collection and control device; 3, walking device [12]; 4, monitoring and image capture device; 5, image capture device—infrared thermal camera; 6, image capture device—visible light camera; 7, visualization interface.

**Figure 2 sensors-24-06385-f002:**
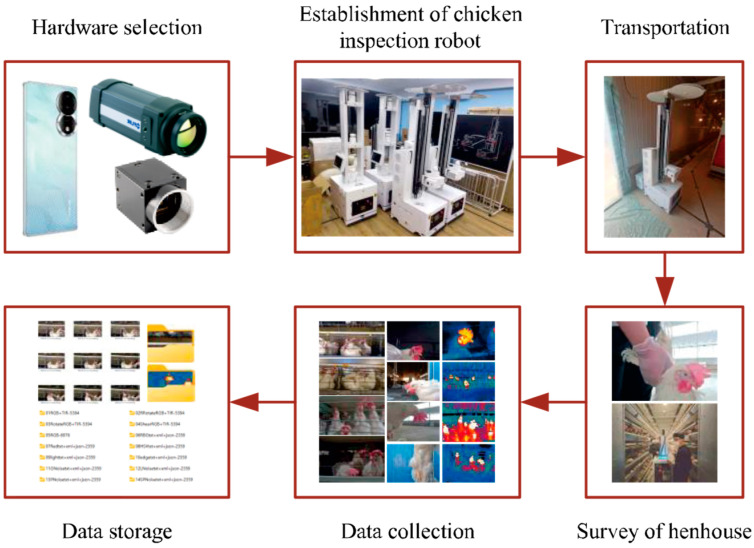
Data collection process. Hardware selection: mobile phones, visible light cameras, infrared thermal imagers, and other peripheral hardware. Establishment of chicken inspection robot: assemble the chicken inspection robot and debug the image acquisition.

**Figure 3 sensors-24-06385-f003:**
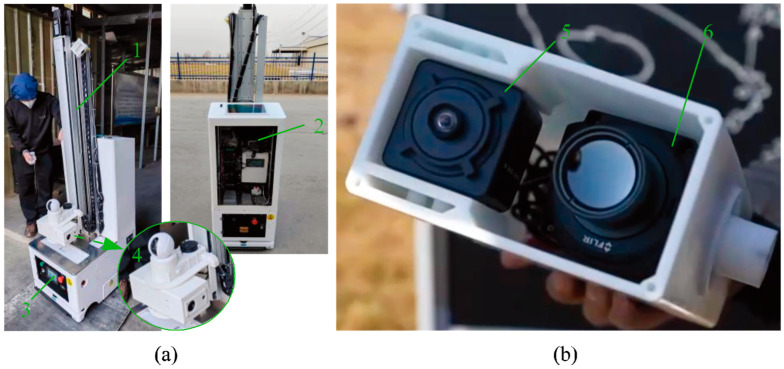
Chicken inspection robot. (**a**) The overall structure of the dead chicken detection robot; (**b**) Image acquisition device. 1, Lifting device of the chicken inspection robot; 2, information collection and control system; 3, locomotion device [12]; 4, monitoring and image capture device; 5, image capture device—infrared thermal camera; 6, image capture device—visible light camera.

**Figure 4 sensors-24-06385-f004:**
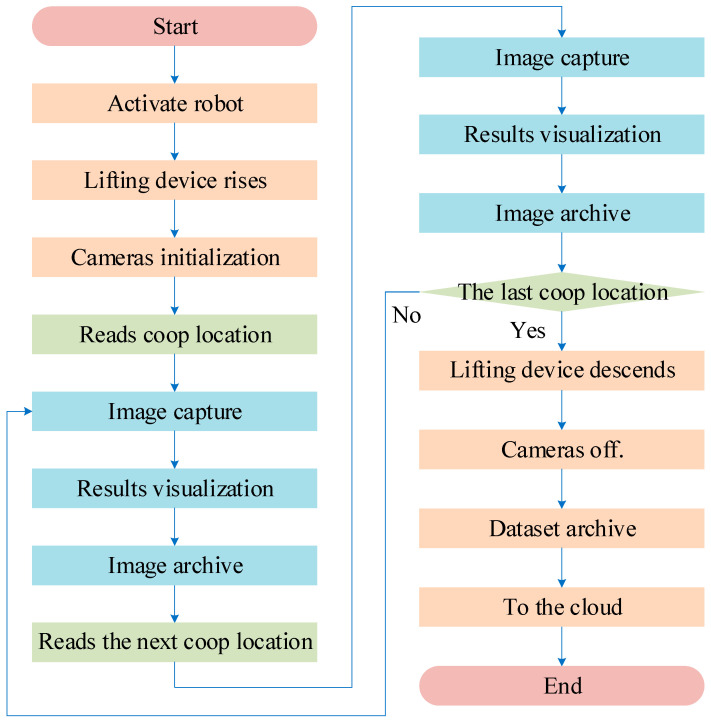
Data collection process flowchart.

**Figure 5 sensors-24-06385-f005:**
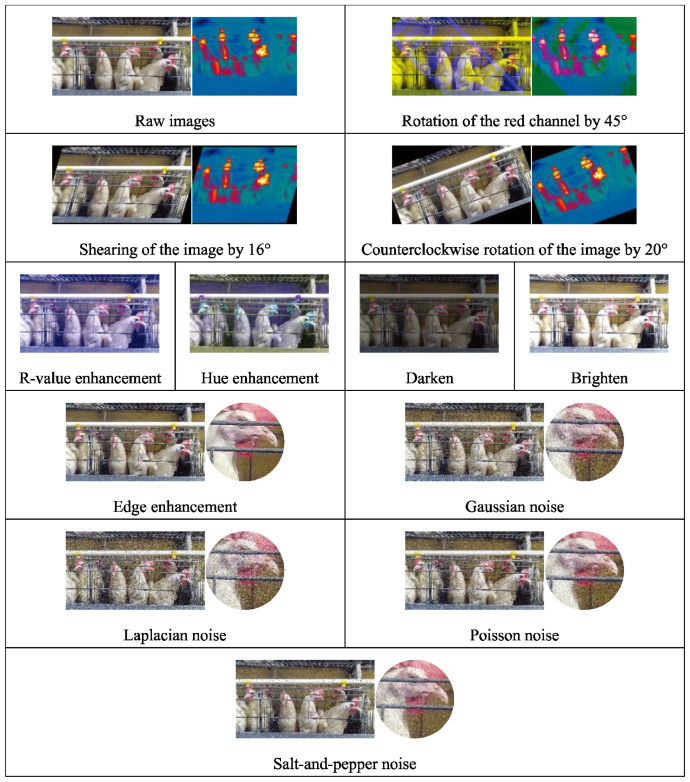
Data augmentation processing.

**Figure 6 sensors-24-06385-f006:**
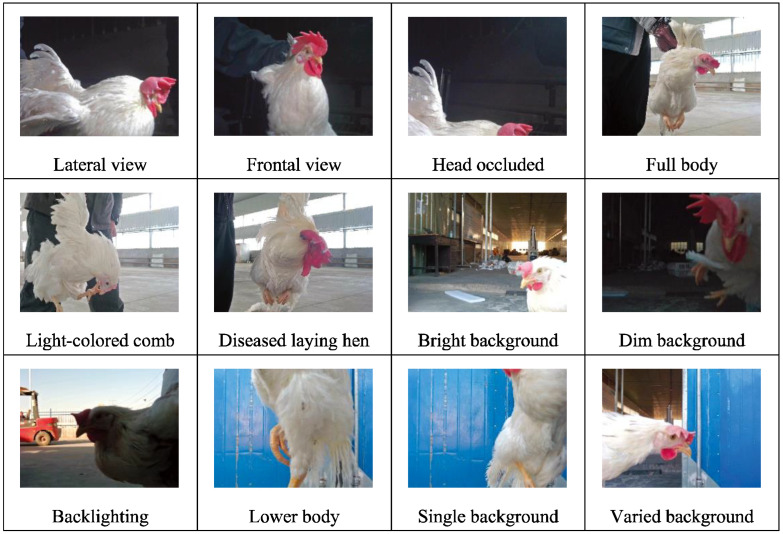
The dataset contains various morphological variations of individual laying hens.

**Figure 7 sensors-24-06385-f007:**
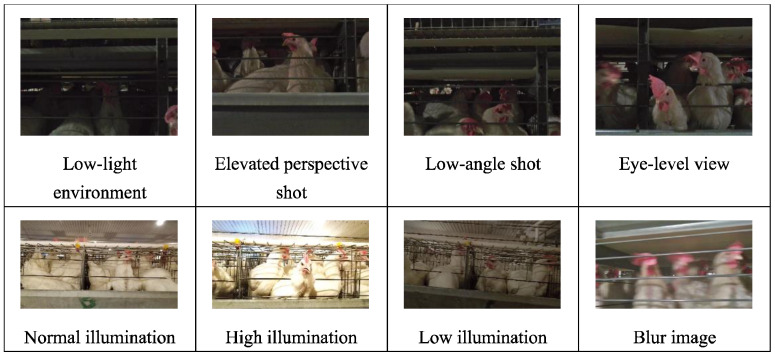
Diverse conditions of caged laying hens in the dataset.

**Figure 8 sensors-24-06385-f008:**
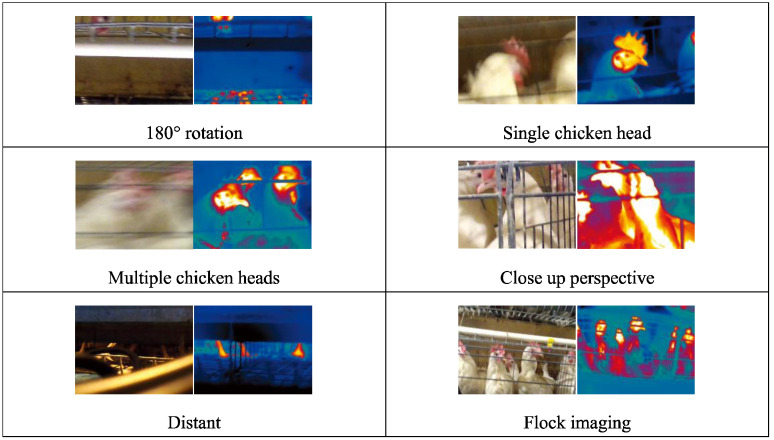
Variability in RGB-TIR image pairings presented in the dataset.

**Figure 9 sensors-24-06385-f009:**
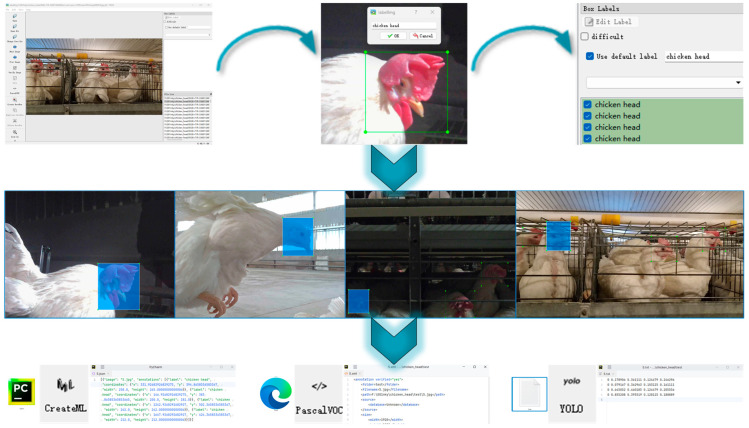
Data annotation process.

**Figure 10 sensors-24-06385-f010:**
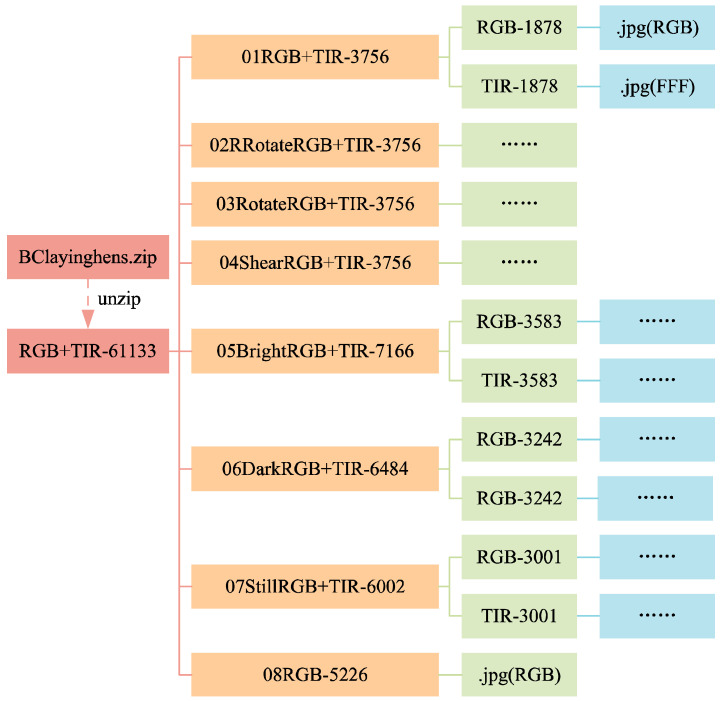
Dataset structure 1.

**Figure 11 sensors-24-06385-f011:**
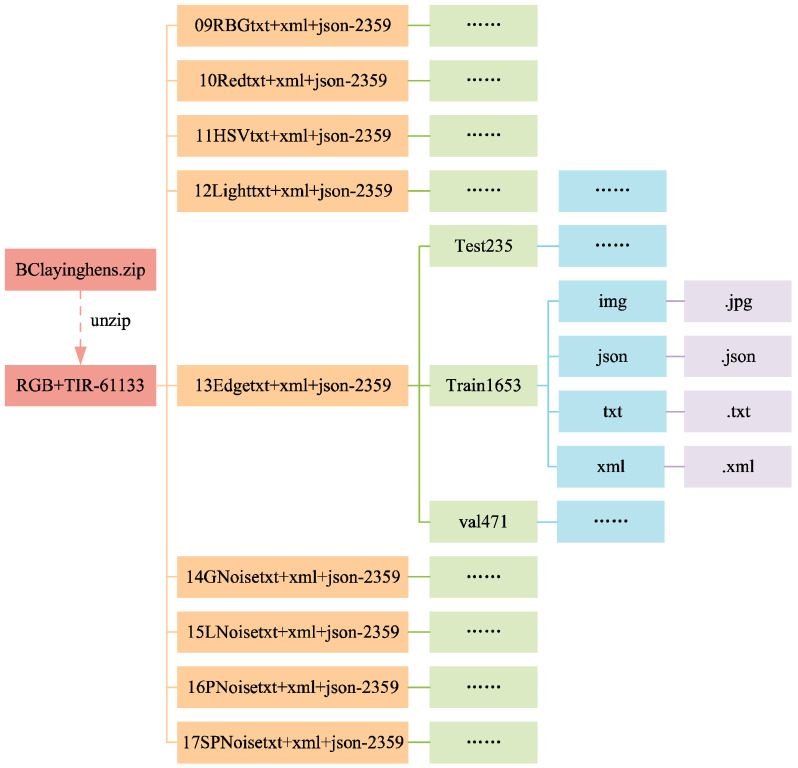
Dataset structure 2.

**Figure 12 sensors-24-06385-f012:**
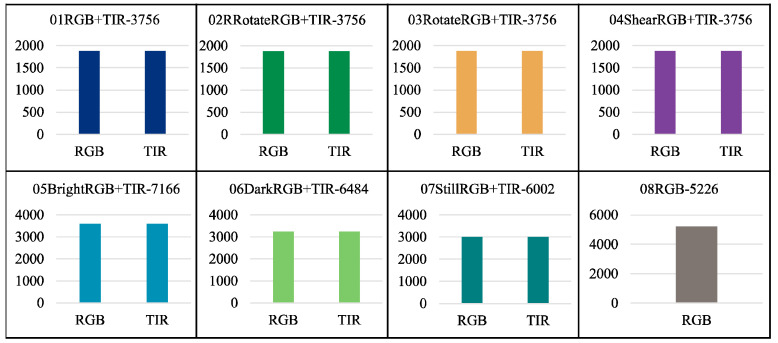
File count of BClayinghens dataset 1.

**Figure 13 sensors-24-06385-f013:**
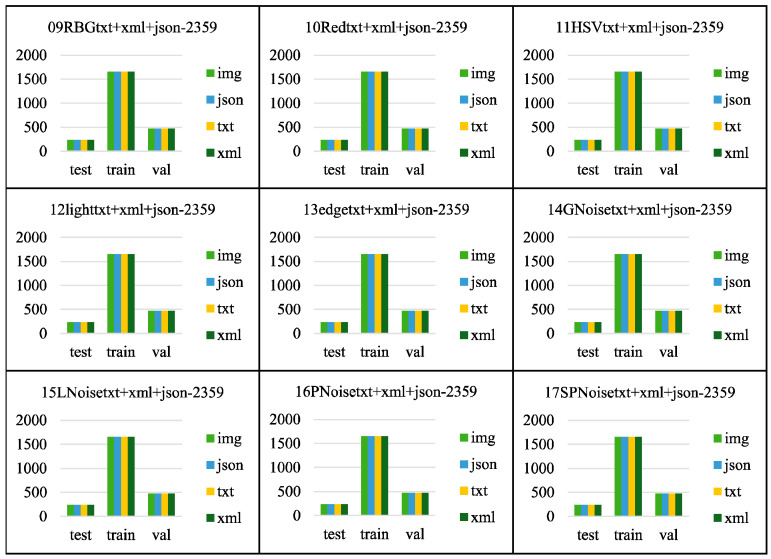
File count of BClayinghens dataset 2.

**Figure 14 sensors-24-06385-f014:**
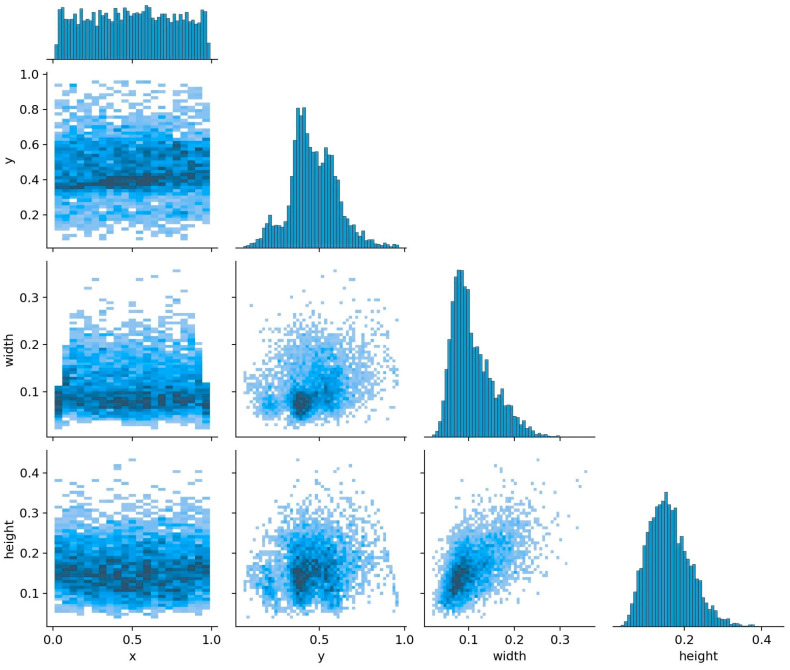
Chicken head detection dataset quality assessment. (x, y) represents the center coordinates of the chicken head target box; width is the width of the chicken head target box; height is the height of the chicken head target box.

**Figure 15 sensors-24-06385-f015:**
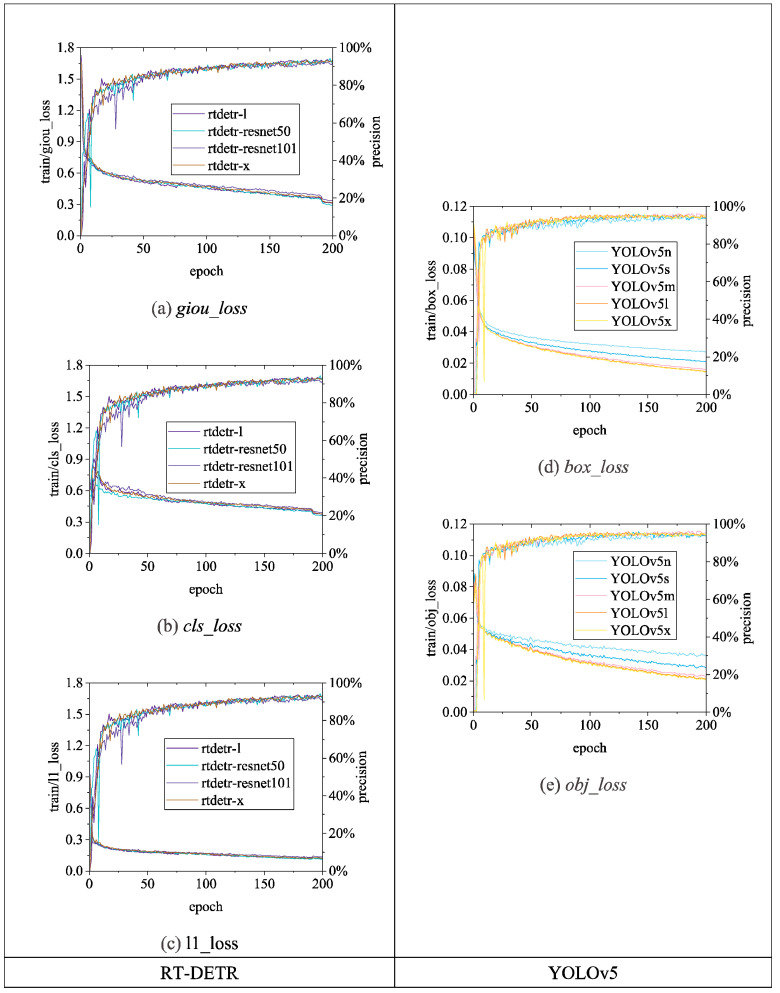
Loss and precision changes of RT-DETR and YOLOv5 chicken head detection models. (**a**) *giou_loss*; (**b**) *cls_loss*; (**c**) l1_*loss*; (**d**) *box_loss*; (**e**) *obj_loss*.

**Figure 16 sensors-24-06385-f016:**
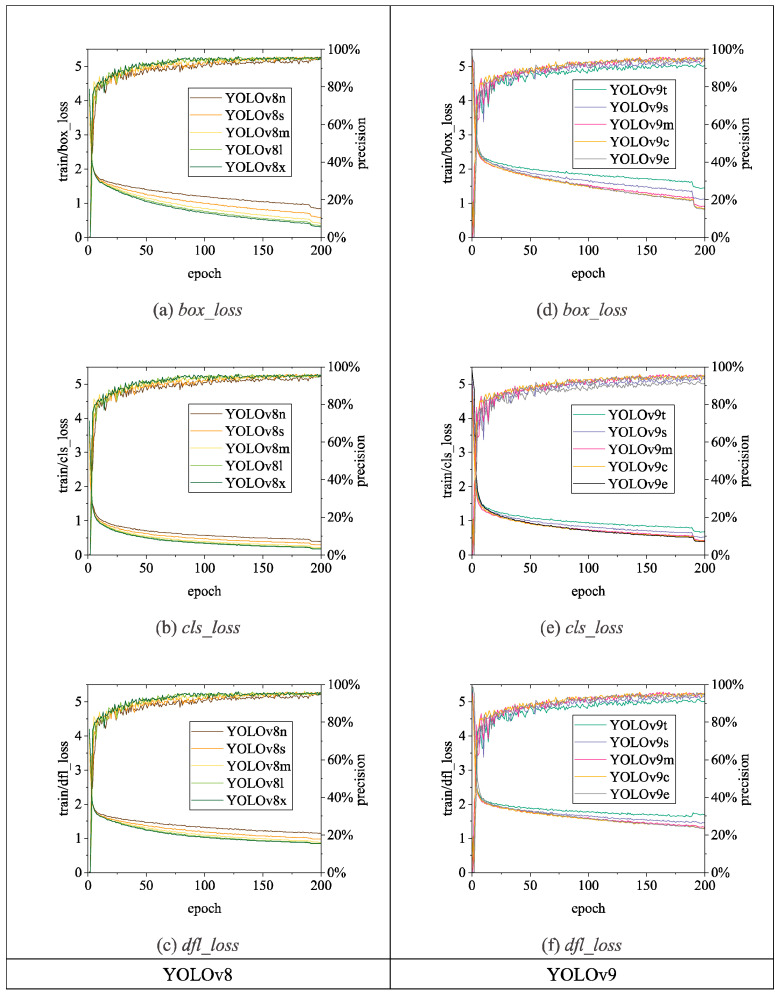
Loss and precision changes of YOLOv8 and YOLOv9 chicken head detection models. (**a**) *box_loss*; (**b**) *cls_loss*; (**c**) *dfl_loss*; (**d**) *box_loss*; (**e**) *cls_loss*; (**f**) *dfl_loss*.

**Figure 17 sensors-24-06385-f017:**
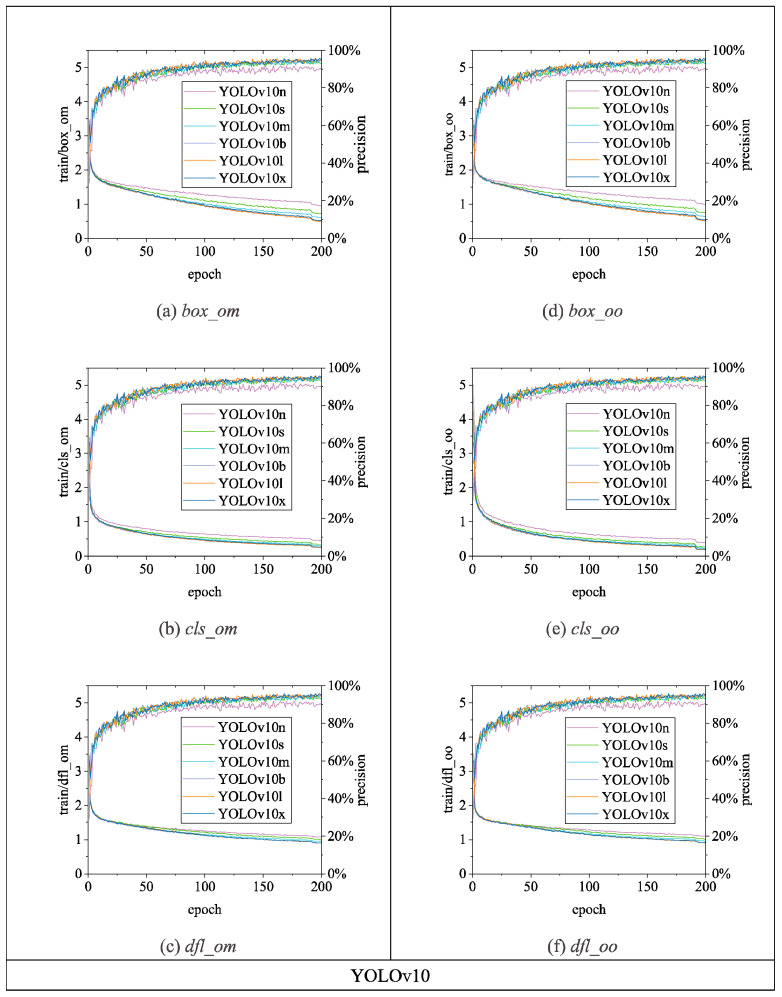
Loss and precision changes of YOLOv10 chicken head detection model. (**a**) *box_om*; (**b**) *cls_om*; (**c**) *dfl_om*; (**d**) *box_oo*; (**e**) *cls_oo*; (**f**) *dfl_oo*.

**Figure 18 sensors-24-06385-f018:**
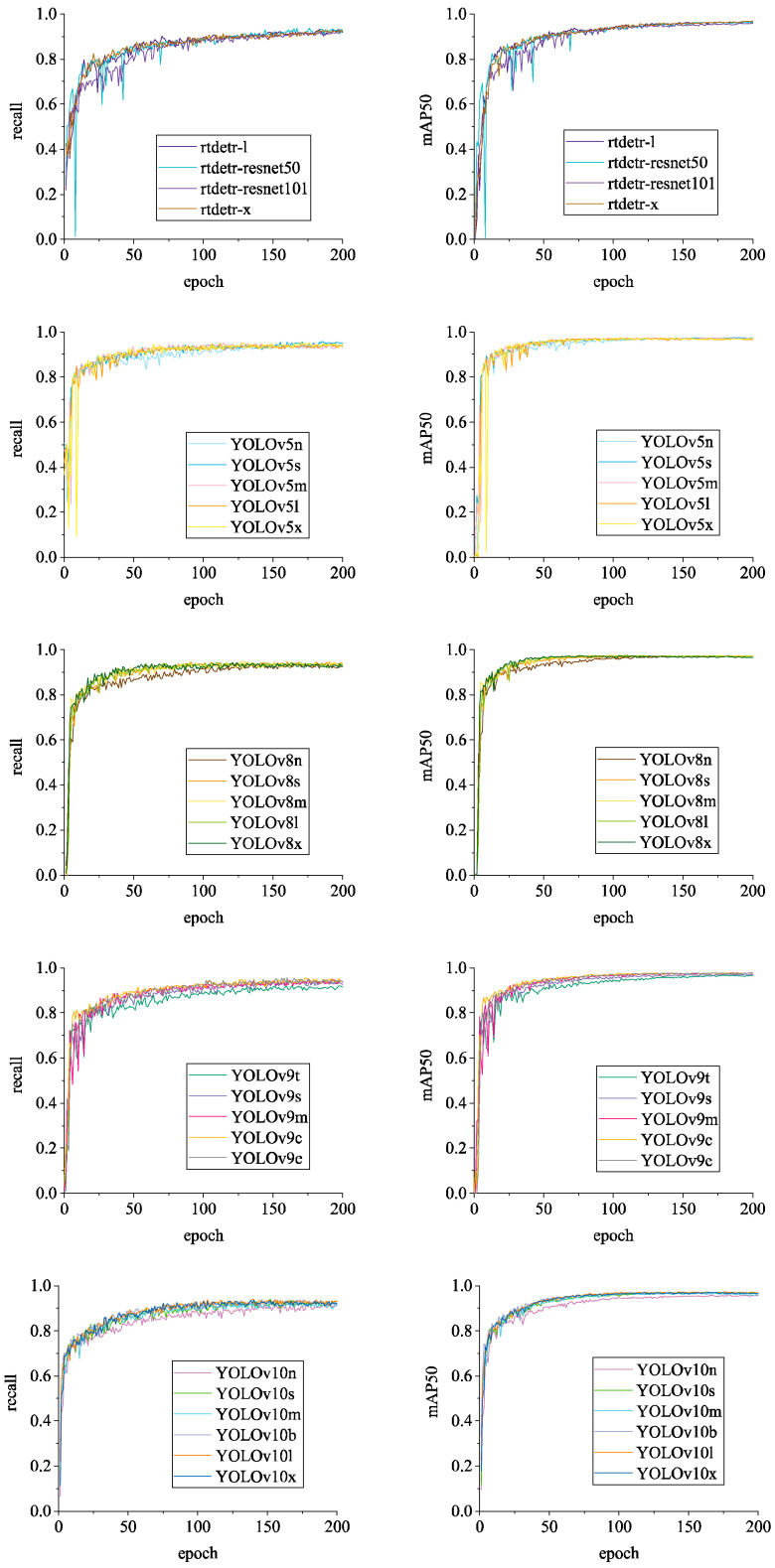
Model Recall and mAP50 change curves.

**Figure 19 sensors-24-06385-f019:**
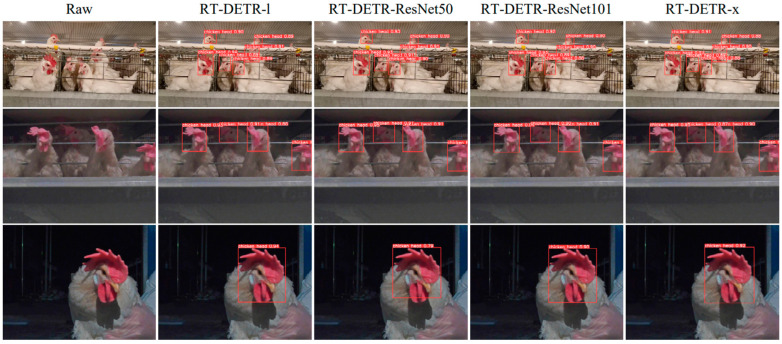
Recognition effect of the RT-DETR object detection algorithm.

**Figure 20 sensors-24-06385-f020:**
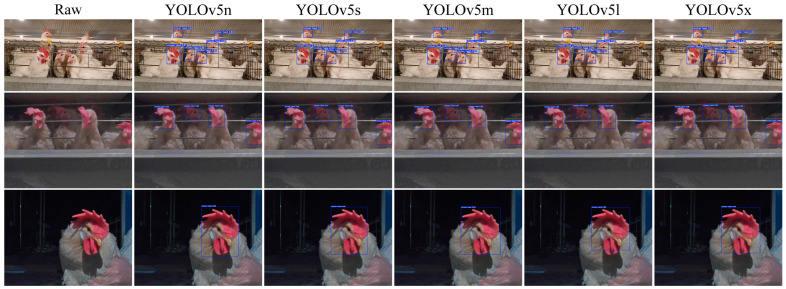
Recognition effect of the YOLOv5 object detection algorithm.

**Figure 21 sensors-24-06385-f021:**
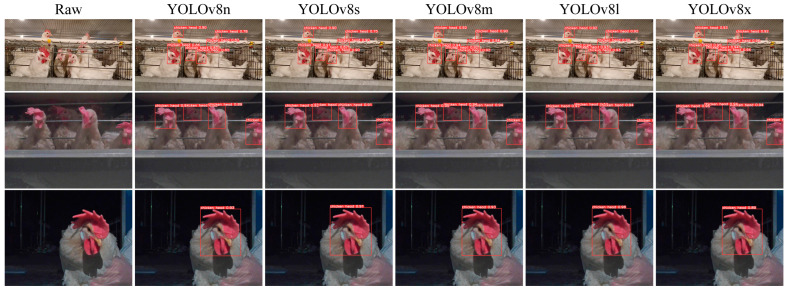
Recognition effect of the YOLOv8 object detection algorithm.

**Figure 22 sensors-24-06385-f022:**
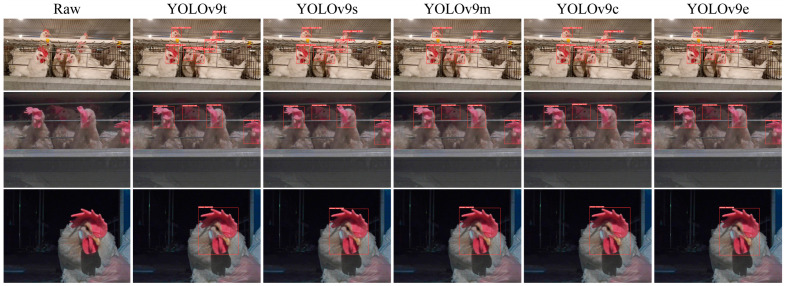
Recognition effect of the YOLOv9 object detection algorithm.

**Figure 23 sensors-24-06385-f023:**
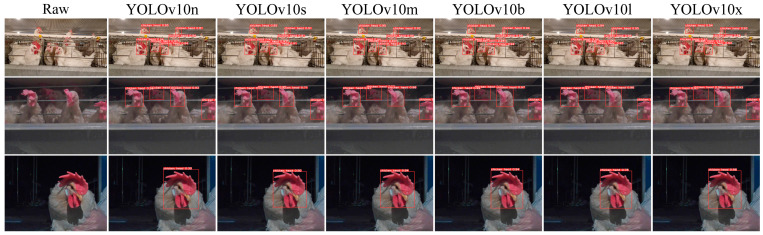
Recognition effect of the YOLOv10 object detection algorithm.

**Figure 24 sensors-24-06385-f024:**
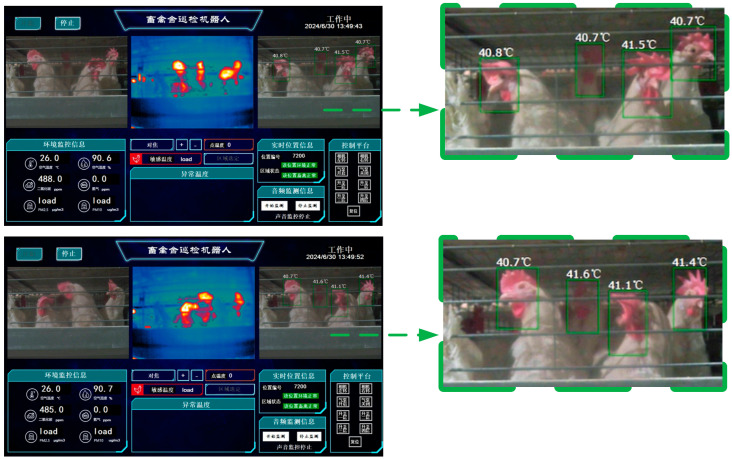
Chicken head temperature recognition effect.

**Table 1 sensors-24-06385-t001:** Technical parameters of the autonomous poultry house inspection robot.

Module	Category	Parameter
Walking Device	Navigation Method	Manual/Automatic Magnetic Navigation
	Charging Method	Self Charging
	Cruise Speed	Ten-speed Adjustment: 3–30 m/min
	Turning Radius	Zero-radius Turning
Control Device	Industrial PC	K345, dual network and six serial ports (6 RS485), 8th generation i7 16 G + 256 G, win10–64 bit
	Touchscreen	11.6-inch (16:9), 1920 × 1080P
	System Information	Windows 10 × 64, Intel(R) Core(TM) i7, 16RAM, Memory 256 G SSD + 1T HDD
	Communication Method	WiFi + 4G
Sensing Device	Visible Light Camera	Details are provided in Table 2
	Thermal Infrared Camera	Details are provided in Table 2
	Gas Sensor	Pump-operated, CO_2_ measuring range [0~5000 ppm, ±5%], NH_3_ measuring range [0~50 ppm, ±0.5 ppm]
	Temperature and Humidity Sensor	Measurement Range [0~50 °C, ±0.2 °C], Measurement Range [0~100%, ±0.3% RH]
Lifting Device	Dimensions	820 mm × 600 mm × [Height adjustable from 514 mm to 2519 mm]
	Lifting Speed	50 mm/s

**Table 2 sensors-24-06385-t002:** Parameters of experimental equipment.

Equipment	Category	Specifications
Smartphone	Camera manufacturer	HONOR (Shenzhen, China)
	Camera model	ANN-AN00
	Resolution	2592 × 1944
Visible light camera on the robot	Camera model	LRCP20680_1080P
	Resolution	1920 × 1080
Infrared thermal camera on robot	Camera manufacturer	FLIR Systems
	Camera model	FLIR A300 9 Hz
	Resolution	320 × 240

**Table 3 sensors-24-06385-t003:** Single experiment data collection plan.

No.	Time	General State of Hens	Capturing Tool	Image Type
1	Day 1, 7 a.m.	Looking around	Smartphone	RGB
2	Day 2, 1 p.m.	Looking around	Robot	RGB + TIR
3	Day 3, 7 p.m.	Resting	Robot	RGB
4	Day 4, 9 p.m.	Resting	Robot	RGB + TIR
5	Day 5, 11 a.m.	Feeding	Smartphone	RGB
6	Day 6, 5 p.m.	Feeding	Robot	RGB + TIR
7	Day 7, 2 p.m.	Manual handling	Smartphone	RGB
8	Day 8, 11: 30 a.m.	Large-scale movement of feed cart	Robot	RGB
9	Day 9, 11 a.m.	Pecking	Smartphone	RGB
10	Day 10, 04: 30 p.m.	Egg collection device movement	Smartphone	RGB
11	Day 11, 9 a.m.	Excited	Robot	RGB + TIR
12	Day 12, 3 p.m.	Excited	Robot	RGB

**Table 4 sensors-24-06385-t004:** Data selection criteria.

Serial Number	Criteria	Content
1	Resolution	2592 × 1944, 1920 × 1080, 320 × 240.
2	Clarity	Delete blurry or out-of-focus images.
3	Environmental Variation	The henhouse has 4 levels, with samples from each level accounting for 25%.
4	Sample Quantity	More than 6 images per cage.
5	Image Format	.jpg
6	Number of Hens per Image	More than 1 hen
7	Action Variation	Looking around, resting, eating, pecking, excitement; each accounting for 20%.

**Table 5 sensors-24-06385-t005:** Specifications table of BClayinghens.

Subject	Dataset of Visible Light and Thermal Infrared Images for Caged Laying Hens, Precision Laying Hen Farming.
Specific Academic Field	Deep learning-based image recognition, counting, health monitoring, and behavioral analysis of caged laying hens.
Data Formats	Raw images, XML annotations, TXT annotations, JSON annotations.
Data Types	Visible light (RGB) images, thermal infrared (TIR) images.
Data Acquisition	A chicken inspection robot together with a smartphone, visible light camera, and infrared thermal imager was used to collect images of laying hens with various poses within a large-scale poultry farm. The collected images comprise RGB and TIR images with resolutions of 2592 × 1944, 1920 × 1080, and 320 × 240, totaling 61,133. The dataset, after compression, is 76.1 GB in size and is available for download in ZIP format.
Data Source Location	Country: China; City: Shengzhou, Hebei; Institution: Shengzhou Xinghuo Livestock and Poultry Professional Cooperative.
Data Accessibility	Repository Name: BClayinghens Direct URL to the Data: https://github.com/maweihong/BClayinghens.git (accessed on 1 September 2024)

**Table 6 sensors-24-06385-t006:** Comparison test results of different models.

*Subject*	*Model Size* *(MB)*	*Parameters* *(M)*	*Computation* *(GFLOPs)*	*Precision* *(%)*	*Recall* *(%)*	*mAP50* *(%)*
*RT-DETR-l*	*66.2*	*32.0*	*103.4*	*93.69*	*93.35*	*96.78*
*RT-DETR-ResNet50*	*86.0*	*41.9*	*125.6*	*94.42*	*93.84*	*96.54*
*RT-DETR-ResNet101*	*124.2*	*60.9*	*186.2*	*92.41*	*92.18*	*95.94*
*RT-DETR-x*	*135.4*	*65.5*	*222.5*	*93.29*	*92.88*	*96.93*
*YOLOv5n*	*3.9*	*1.9*	*4.5*	*94.52*	*94.92*	*97.52*
*YOLOv5s*	*14.4*	*7.2*	*16.5*	*95.98*	*95.82*	*97.54*
*YOLOv5m*	*42.2*	*21.2*	*49.0*	*96.27*	*95.13*	*97.59*
*YOLOv5l*	*92.8*	*46.5*	*109.1*	*95.92*	*94.87*	*97.27*
*YOLOv5x*	*173.1*	*86.7*	*205.7*	*95.59*	*95.23*	*97.35*
*YOLOv8n*	*6.3*	*3.2*	*8.7*	*95.06*	*94.16*	*97.51*
*YOLOv8s*	*22.5*	*11.2*	*28.6*	*95.97*	*94.83*	*97.38*
*YOLOv8m*	*52.1*	*25.9*	*78.9*	*96.48*	*94.89*	*97.71*
*YOLOv8l*	*88.7*	*43.7*	*165.2*	*95.98*	*94.42*	*97.55*
*YOLOv8x*	*136.7*	*68.2*	*257.8*	*96.28*	*94.29*	*97.43*
*YOLOv9t*	*6.4*	*2.0*	*7.7*	*92.24*	*92.12*	*96.68*
*YOLOv9s*	*20.3*	*7.1*	*26.4*	*94.07*	*94.69*	*97.21*
*YOLOv9m*	*66.2*	*20.0*	*76.3*	*95.97*	*95.09*	*97.92*
*YOLOv9c*	*102.8*	*25.3*	*102.1*	*95.96*	*95.54*	*97.81*
*YOLOv9e*	*140.0*	*57.3*	*189.0*	*95.85*	*95.73*	*97.87*
*YOLOv10n*	*5.8*	*2.3*	*6.7*	*91.87*	*92.54*	*95.94*
*YOLOv10s*	*16.6*	*7.2*	*21.6*	*95.18*	*94.02*	*97.24*
*YOLOv10m*	*33.5*	*15.4*	*59.1*	*94.62*	*92.54*	*96.81*
*YOLOv10b*	*41.5*	*19.1*	*92.0*	*95.62*	*93.84*	*97.13*
*YOLOv10l*	*52.2*	*24.4*	*120.3*	*95.94*	*94.04*	*97.32*
*YOLOv10x*	*64.1*	*29.5*	*160.4*	*95.74*	*93.67*	*97.17*

## Data Availability

Direct URL to the data: https://github.com/maweihong/BClayinghens.git (accessed on 1 September 2024).

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
