# Peer review of "A Dataset of Visible Light and Thermal Infrared Images for Health Monitoring of Caged Laying Hens in Large-Scale Farming"

_sensors, 2024, doi:10.3390/s24196385_

Round 1

Reviewer 1 Report

Comments and Suggestions for Authors

The manuscript submitted for review provides sufficient background and includes all references. However, a precise formulation of the aim of the manuscript and the specific tasks that the authors set themselves is missing.

The study design is appropriate and comprehensive. However, the discussion of the presented results is lacking too. The publication would be more in-depth if the authors compared their results with the capabilities of other methods for recording the behaviour and temperature of birds, as presented in the introduction of the manuscript. 

Comments on the Quality of English Language

I think the English language used in the manuscript is good. 

Reviewer 2 Report

Comments and Suggestions for Authors

This manuscript constructed a dataset named "BClayinghens" containing visible light and thermal infrared images of caged laying hens in large-scale farming. The dataset aims to enable comprehensive evaluation of the hens' health, behavior, and population counts.

The dataset appears to be one of the first of its kind, providing a large collection of visible light and thermal infrared images of caged laying hens, which can be valuable for developing and training machine learning models for various applications in precision poultry farming, such as  disease detection and thermal control.

My major concerns would be:

1. The detailed information on the data collection process is unclear, such as the duration of data collection, the number of farms involved, and the specific criteria used for selecting the images.

2. This manuscript does not provide any evaluation, such as abenchmark of the dataset using machine learning models or specific tasks, which would have demonstrated the dataset's effectiveness and potential applications.

3. This manuscript does not discuss any potential ethical concerns related to the collection and use of animal data or the implications of using this dataset for automated monitoring and management of caged laying hens.

Based on the provided information, I would recommend reconsidering the paper after major revisions. The dataset appears to be a valuable contribution to the field of precision poultry farming, and the authors have made efforts to ensure data diversity and quality. However, the authors should address above points first.

In addition, there are some minor suggestions:

1. try provide details on the data collection equipment, experimental methods with figures.

2. The references appear to be appropriate and try cover relevant literature in the field of precision poultry farming and computer vision applications

3. The authors could consider providing more specific details on the data annotation process, such as the number of annotators involved, the annotation guidelines, and any measures taken to ensure annotation quality and consistency.

Round 2

Reviewer 2 Report

Comments and Suggestions for Authors

The manuscript was revised based on the review comments.